# Blood Cerebrospinal Fluid Barrier Function Disturbance Can Be Followed by Amyloid-β Accumulation

**DOI:** 10.3390/jcm11206118

**Published:** 2022-10-17

**Authors:** Yuji Suzuki, Yukimi Nakamura, Hironaka Igarashi

**Affiliations:** Center for Integrated Human Brain Science, Brain Research Institute, University of Niigata, Niigata 951-8585, Japan

**Keywords:** interstitial flow, amyloid-β accumulation, [^15^O]H_2_O PET, [^18^F]flutemetamol PET, blood–cerebrospinal fluid barrier (BCSFB), glymphatic system

## Abstract

Background: Elucidation of the mechanism of amyloid-β accumulation plays an important role in therapeutic strategies for Alzheimer’s disease (AD). The aim of this study is to elucidate the relationship between the function of the blood–cerebrospinal fluid barrier (BCSFB) and the clearance of amyloid-β (Aβ). Methods: Twenty-five normal older adult volunteers (60–81 years old) participated in this PET study for clarifying the relationship between interstitial water flow and Aβ accumulation. Water dynamics were analyzed using two indices in [^15^O]H_2_O PET, the influx ratio (IR) and drain rate (DR), and Aβ accumulation was assessed qualitatively by [^18^F]flutemetamol PET. Results: [^15^O]H_2_O PET examinations conducted initially and after 2 years showed no significant changes in both indices over the 2-year period (IR: 1.03 ± 0.21 and 1.02 ± 0.20, DR: 1.74 ± 0.43 and 1.67 ± 0.47, respectively). In [^18^F]flutemetamol PET, on the other hand, one of the 25 participants showed positive results and two showed positive changes after 2 years. In these three participants, the two indices of water dynamics showed low values at both periods (IR: 0.60 ± 0.15 and 0.60 ± 0.13, DR: 1.24 ± 0.12 and 1.11 ± 0.10). Conclusions: Our results indicated that BCSFB function disturbances could be followed by Aβ accumulation, because the reduced interstitial flow preceded amyloid accumulation in the positive-change subjects, and amyloid accumulation was not observed in the older adults with sufficiently high values for the two indices. We believe that further elucidation of interstitial water flow will be the key to developing therapeutic strategies for AD, especially with regard to prevention.

## 1. Introduction

Disturbances in the proper clearance of amyloid-β (Aβ) because of a negative balance between its clearance and production have been implicated to play a significant role in senile plaque formation and ultimately the pathogenesis of Alzheimer’s disease (AD) [1]. The clearance of Aβ involves the blood–cerebrospinal fluid barrier (BCSFB), which functions to prevent the passage of most blood-borne substances into the brain while selectively permitting the passage of specific substances into the brain and facilitating the removal of brain metabolites and metabolic products such as Aβ [2,3].

The interstitial water flow, which is suggested to be regulated by aquaporin 4 (AQP4) expressed in the terminal foot of astrocytes [4], plays an important role in the removal function [5,6,7]. The paravascular pathway facilitates CSF flow through the brain parenchyma and the clearance of interstitial solutes, including Aβ. The flow has been postulated to play a role similar to systemic lymphatics in the brain, and this system is now widely referred to as the glymphatic system [8].

The glymphatic system has been proposed to remove waste products from the cerebrospinal fluid (CSF) through the interstitial water flow [8,9,10,11]. Recent studies have reported altered AQP4 expression and localization in the human brain with aging [12]. These alterations are most likely caused by disturbances in proper clearance. Our water dynamics analysis, performed using [^15^O]H_2_O positron emission tomography (PET), showed that water influx into the CSF tends to decrease with age and that patients with AD show a significant reduction in interstitial flow [13]. These results indicate that decreased interstitial water flow is one of the risk factors for AD. However, it remains unclear whether the observed decrease in interstitial water flow is a primary change associated with aging or a secondary change caused by Aβ deposition in the brain. 

We believe that elucidating the relationship between Aβ deposition and interstitial water flow, which is central to the clearance function, is important in understanding the pathogenesis in the very early stages of AD and, ultimately, the therapeutic strategies for AD. Therefore, we conducted a 2-year follow-up study using both [^15^O]H_2_O and amyloid PET ([^18^F]flutemetamol PET) to analyze Aβ accumulation and changes in interstitial water flow and aimed to clarify this relationship.

## 2. Materials and Methods

### 2.1. Participants

Twenty-five normal older adult volunteers (13 males and 12 females; age, 60–81 years) participated in this follow-up study and were evaluated eventually (Figure 1). 

Participant recruitment was completed between September 2015 and March 2017. All participants were free of any significant medical conditions, such as chronic heart, kidney, and pulmonary diseases, and were not taking any prescribed, over the counter, or herbal medications. The volunteers were assessed to have no functional or cognitive impairment (Mini-Mental State Examination [MMSE] score ≥ 28) and had no neurological disease. In compliance with the guidelines of the Institutional Review Board of the University of Niigata, the study protocol was explained in detail to all potential participants or their proxies where appropriate, and written informed consent was obtained from all participants. All participants were provided with contact names and telephone numbers in the event of any adverse event related to the study and were followed up within one month of the study to further confirm the absence or occurrence of study-related adverse events that were not otherwise self-reported by the study participant or proxy to the study coordinator. This follow-up project was conducted according to the human research guidelines of the Institutional Review Board of the University of Niigata, under the approval of the research ethics committee (approval numbers: 2015–2589), and the [^15^O]H_2_O PET study was registered at the UMIN Clinical Trials Registry as UMIN000011939. Each participant underwent both [^15^O]H_2_O and [^18^F]flutemetamol PET imaging procedures two weeks apart, and then underwent the same PET imaging procedures in a follow-up assessment performed two years later.

### 2.2. PET Imaging

PET imaging was performed using a combined PET/CT scanner (Discovery ST Elite, GE Healthcare, Schenectady NY, USA) with a 15-cm field of view (FOV) positioned in the region of the cerebrum. For correcting photon attenuation, a low-dose CT scan was acquired in helical mode with the following parameters: 120 kV, 50 mA, 0.8 s per tube rotation, slice thickness of 3.75 mm with intervals of 3.27 mm, pitch of 0.875, and a table speed of 17.5 mm/rotation. During the scanning procedure, the participant’s head was rested on a foam-cushioned headrest, and a head strap was used to minimize head movement. All PET emission scans were normalized for detector inhomogeneity and corrected for random coincidences, dead time, scattered radiation, and photon attenuation. These scans were reconstructed using three-dimensional ordered-subset expectation maximization (3D-OSEM) with two iterations and 28 subsets to obtain superior visual quality images, allowing manual definition of regions of interest (ROIs). For the reconstruction algorithms, the data were collected in a 128 × 128 × 47 matrix with a voxel size of 2.0 × 2.0 × 3.27 mm.

#### 2.2.1. [^15^O]H_2_O PET

A 1000-MBq [^15^O]H_2_O synthesized online was injected intravenously using an automatic water injection system, followed immediately by a 10-mL saline flush at the speed of 1 mL/s (AM WR01; JFE Technos, Yokohama, Japan). After starting the injection, PET emission data were promptly acquired over 20 min in three-dimensional list mode with a 25.6-cm axial FOV and sorted into 47-time frames (18 × 10 s, 24 × 30 s, 5 × 60 s). The CT and PET image data were transferred to a Xeleris 3.1 workstation (GE Healthcare) for PET data analysis. Manually defined ROIs (lateral and third ventricles, cortex of the frontal and occipital lobes) utilizing CT and [^15^O]H_2_O PET images were drawn using volumetrix MI on a Xeleris 3.1 workstation (Figure 2). 

The tissue activity concentration in each ROI was expressed as the standardized uptake value (SUV, g/mL), corrected for the participant’s body weight and administered dose of radioactivity. Each tissue time activity concentration was fitted from the peak point in the cortex and from the start point in the ventricle to the following exponential curve by using SigmaPlot version 14.5 (Systat Software Inc., Chicago, IL, USA).
y(t) = y_0_ + a*e*^−bt^
where y_0_ and b in each tissue were the implied tissue baseline SUV and in/out-flow pace, respectively (Figure 3A). To assess interstitial flow, we focused on two indices, namely, the ventricle and cortical y_0_ ratio as the indicator of water influx into the CSF space from the cortex and the cortical b as the speed of drainage flow from the cortex. As shown in Figure 3B, the former was defined as the influx ratio (IR) = y_0(ventricle)**/**_y_0(cortex_, representing the ratio of water flowing into the CSF space from the cortex, with higher values indicating a larger water flow. The latter was defined as the drain rate (DR) = b_(cortex)_ × 10^3^, representing the pace of water drainage into the CSF space from the cortex, with higher values indicating a faster flow speed. As a control, we used previously reported data from 10 young individuals (21–30 years old) and 10 AD patients (59–84 years old).^15^ Statistical analyses, *t*-tests, and paired-sample *t*-tests were performed using SPSS version 19.0 (IBM, Chicago, IL, USA), and *p*-values less than 0.01 were regarded as statistically significant.

#### 2.2.2. [^18^F]flutemetamol (Vizamyl^®^) PET

An intravenous bolus injection of 162–221 MBq [^18^F]flutemetamol (3.0–3.3 MBq/kg), produced using an automated synthesizer (FASTlab; GE Healthcare, Schenectady, NY, USA) was administered, and a 20-min PET scan in 3-dimensional statistic mode was started after 90 min in accordance with the imaging acquisition guidelines. The PET images scaled to 90% of the pons were visually assessed as either amyloid-positive or amyloid-negative on the basis of the training program instructions provided by GE Healthcare for the interpretation of [^18^F]flutemetamol images. A negative scan shows more radioactivity in the white matter than in the gray matter. Conversely, a positive scan shows gray matter radioactivity as intense as or exceeding that in the adjacent white matter in at least one of the five key regions (the posterior cingulate gyrus and precuneus, frontal cortex, lateral temporal cortex, parietal cortex, and striatum) (Figure 4). 

## 3. Results

The results are summarized in Table 1. In this study, participants were followed up at intervals of 2 years (range of interval periods: 2.0–2.8 years, mean ± SD: 2.2 ± 0.2 years). In the [^15^O]H_2_O PET examinations performed at the start and after 2 years, IR was 1.03 ± 0.21 and 1.02 ± 0.20, respectively, and DR was 1.74 ± 0.43 and 1.67 ± 0.47, respectively; both indices showed no significant changes over the 2-year period in the paired-sample *t*-test. On the other hand, as shown in Figure 4, the values for both indices were significantly lower than those in young controls (blue open circles and bars; IR = 1.38 ± 0.08 and DR = 2.12 ± 0.32) and significantly higher than those of the AD patients (red open circles and bars; IR = 0.73 ± 0.09 and DR = 0.86 ± 0.17). 

At the initial [^18^F]flutemetamol PET examination, one of the 25 participants showed positive results and at the 2-year follow-up PET, two participants showed positive changes. All the other 22 participants showed negative results in both periods. The relationship between [^15^O]H_2_O PET and [^18^F]flutemetamol PET is shown in Figure 5. Red dots show the IR and DR results for the participant showing positive results in PET examinations conducted in both periods, and yellow dots show the results for the two participants who showed positive changes on follow-up PET. These three participants showed significantly low IR and DR values in both periods (IR, 0.60 ± 0.15 and 0.60 ± 0.13; DR, 1.24 ± 0.12 and 1.11 ± 0.10).

## 4. Discussion

Blood cerebrospinal fluid barrier function disturbances may be followed by Aβ accumulation. In the present study, we performed a follow-up evaluation using [^15^O]H_2_O and [^18^F]flutemetamol PET to analyze the relationship between Aβ accumulation and interstitial flow. The two indices for interstitial flow, IR and ER, did not change significantly over the two-year period in the older adults without cognitive impairment, but their ranges varied widely and were significantly lower than those in the younger volunteers and significantly higher than those in the AD patients.

In the participants showing amyloid PET-positive results and positive changes, the values for the two indices were clearly low at both periods, in comparison with those in AD patients, and the results suggested that the clearance function caused by interstitial flow would be impaired even before Aβ accumulation in those with positive changes. On the other hand, many participants showed negative amyloid PET images, even though these two indices were similarly low. These different results indicate that Aβ accumulation can be caused by an imbalance between production and excretion, and that reduced interstitial flow itself is not necessarily a sufficient factor for Aβ accumulation. On the other hand, since Aβ accumulation was not observed in other participants with sufficiently high values for both indices, in theory, this implied that the reduction in interstitial flow was a necessary condition for Aβ accumulation. Although the results were from an exploratory small-scale study, we believe these are a first step in suggesting that adequate interstitial flow could potentially prevent β-amyloid accumulation. 

We directly observed the dynamics of water by using [^15^O]H_2_O PET to analyze water influx into the CSF space from the cortex. One of the problems of quantitative analysis using PET is that even if the dose is fixed and the signal is corrected by SUV, analysis of each voxel value as an absolute value is difficult because of the differences in distribution among individuals. Therefore, it is necessary to select indices that are not affected by differences in distribution among individuals. Since the purpose of this study was to evaluate the water influx into the CSF space from the cortex, we defined two indices, IR (the ratio of water flowing into the CSF space from the cortex) and DR (the pace of water flowing into the CSF space from the cortex). These two indices were not affected by individual differences in distribution because IR uses the ratio, whereas DR evaluates the form of decay. On the basis of the above findings, these two indices were considered suitable for quantitative analysis of water dynamics.

Our report may help establish methods for the treatment and prevention of AD. In contrast to the circulatory system, which is controlled by AQP-1 [14], the glymphatic system is suggested to be mediated by AQP-4, which is abundantly expressed in the perivascular endfeet of astrocytes. Accordingly, brain interstitial flow, which plays a role equivalent to the systemic lymphatic system, is now considered to be an AQP-4-dependent system [15]. The drainage of Aβ through the interstitial flow into the CSF is essential to maintain proper homeostasis between Aβ production and clearance. Disruption of β-amyloid homeostasis may play an important role in the development of senile plaques. In other words, maintaining the function of interstitial flow may be one of the most promising ways to counteract the development of AD.

As mentioned above, AQP4 is considered a key component of this interstitial flow; nevertheless, the precise interactions with AQP4 have not been fully elucidated. Furthermore, the factors that influence AQP4 expression and function remain largely unknown. A recent immunohistochemical study to evaluate AQP4 expression reported that its expression and distribution change with AD and aging [12]. The altered AQP4 expression and the loss of perivascular AQP4 localization with aging may be a factor that renders the aging brain vulnerable to the misaggregation of Aβ. Further analysis is needed to elucidate the relationship between AQP4 and interstitial flow, and ideally, alterations in expression, distribution, and function should be analyzed in vivo over time. One of the solutions to address these requirements is aquaporin PET ([^11^C]TGN-020 PET) [16,17]. The future challenge using these in vivo imaging techniques is to evaluate the relationship between AQP4 and various neurodegenerative conditions, such as AD, on the basis of interstitial flow over time.

However, this study has some limitations. The methods for evaluating interstitial flow in the brain have not been established yet. We attempted to evaluate the dynamics of water directly using labeled water ([^15^O]H_2_O). In the analysis, IR and DR were newly developed as indices to evaluate changes in interstitial flow into the CSF spaces from the cortex. Physiological factors associated with aging (e.g., brain atrophy, decrease in cerebral blood flow) can have a significant impact on interstitial flow. We assessed water dynamics based on the hypothesis that the resulting reduction in interstitial flow, including these various factors, would be a risk for Aβ accumulation. However, these factors may contribute to and influence the measurement of the two indices, IR and DR. Therefore, to accurately evaluate changes in interstitial flow, the need for corrections to these factors and analytical models must be considered.

In addition, the sample size and observation period of the present study are not sufficient to determine the relationship between interstitial flow and Aβ accumulation. Since only two participants had positive amyloid PET changes, the present results only strongly suggest that the decrease in interstitial flow may precede the appearance of Aβ accumulation. Therefore, we believe that accurate analysis of the relationship between the two can be achieved by conducting observations on a large scale and for a sufficient observation period.

## 5. Conclusions

Our results suggest that disturbances in the proper clearance of Aβ by interstitial flow into the CSF can play a significant role in senile plaque formation and ultimately the pathogenesis of AD. Although this is a small-scale exploratory study, we believe that the results are worth considering to elucidate the very early pathogenesis of AD. In addition, interstitial water flow has also been reported to affect the clearance of tau, focusing additional attention on the therapeutic strategies for AD [18]. Furthermore, the findings indicate the need to evaluate and analyze the various physiological pathological, and histological factors that affect interstitial water flow and impair AQP4 function. We are convinced that further elucidation of these factors will yield important clues to therapeutic strategies for AD, especially for the prevention of this condition.

## Figures and Tables

**Figure 1 jcm-11-06118-f001:**
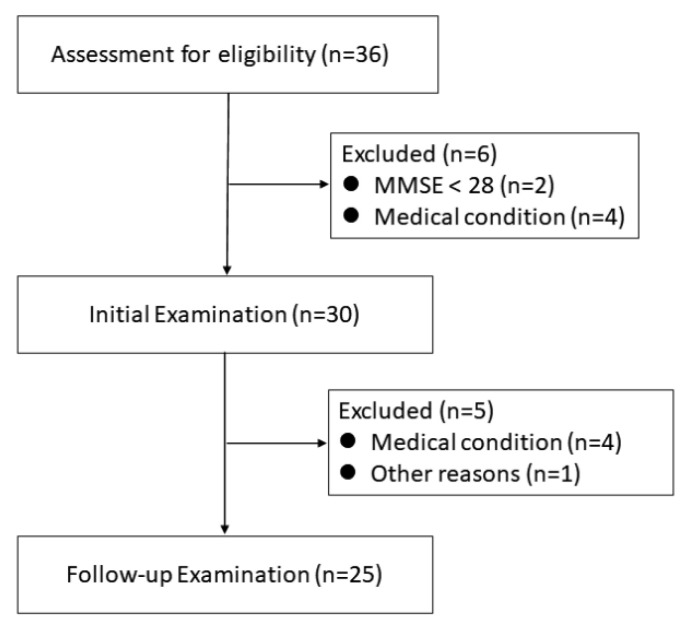
Flow diagram providing details of participant enrolment.

**Figure 2 jcm-11-06118-f002:**
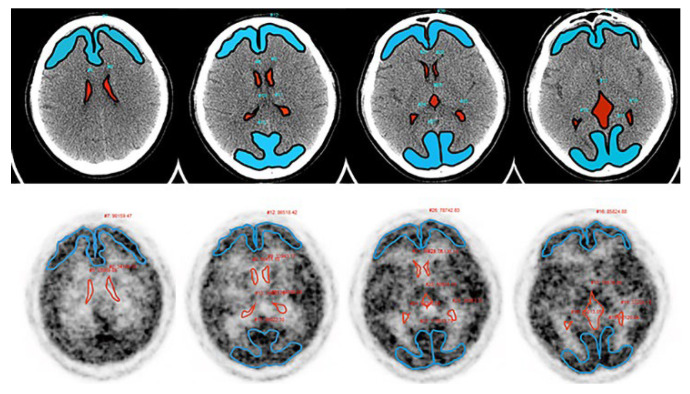
The ROIs for analyzing the interstitial water flow were defined utilizing CT (**upper**) and [^15^O]H_2_O PET (**lower**) images on a Xeleris 3.1 workstation. [^15^O]H_2_O PET images in 60–120 s after the administration were reconstructed to confirm the cortex, which had high uptake regions. Blue areas: frontal and occipital cortex. Red areas: lateral and third ventricle.

**Figure 3 jcm-11-06118-f003:**
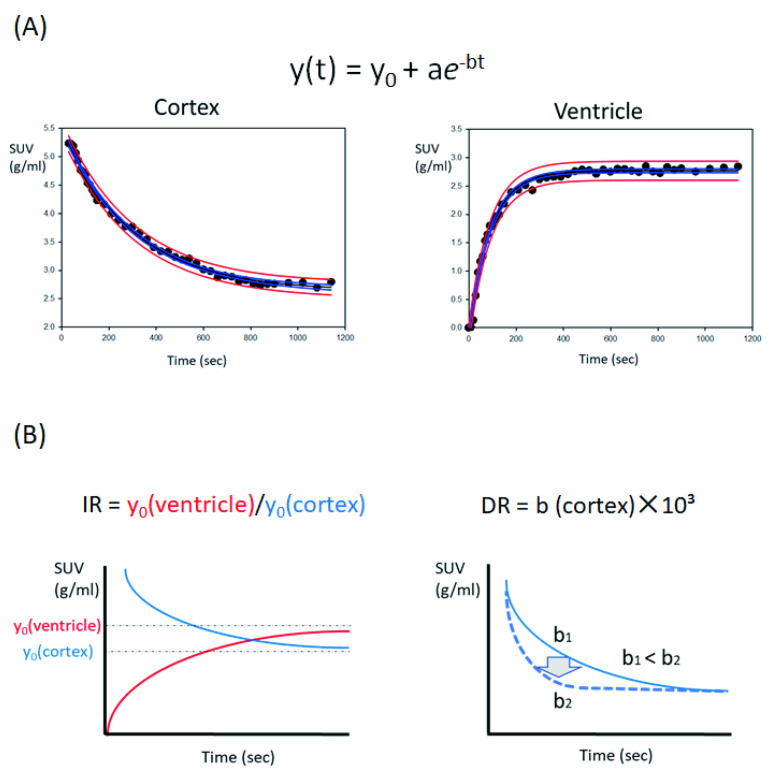
Data analysis in [^15^O]H_2_O PET. (**A**) Time activity concentration presented as the SUV (g/mL) in the cortex and ventricle. The data were fitted to an exponential curve, y(t) = y_0_ + a*e*^−bt^. The blue and red lines represent the 95% confidence and prediction band. The parameters y_0_ and b are the implied tissue baseline SUV and flow pace, respectively. (**B**) Two indices were used for assessing interstitial flow: influx ratio (IR) and drain rate (DR). The two indices were defined as follows: IR = y_0(ventricle)**/**_y_0(cortex)_ and DR = b_(cortex)_ ×10^3^, and the blue and red lines in the schematic figure represent the cortex- and ventricle-fitted curves. IR is the index for the flow of water into the CSF space from the cortex, with a higher value indicating a larger water flow. DR is the index for the rate of water drainage into the CSF space from the cortex, with a larger value indicating a higher flow speed. When b_2_ (dotted line) is larger than b_1_ (solid line), as shown in the figure, the flow speed of b_2_ is higher than that of b_1_.

**Figure 4 jcm-11-06118-f004:**
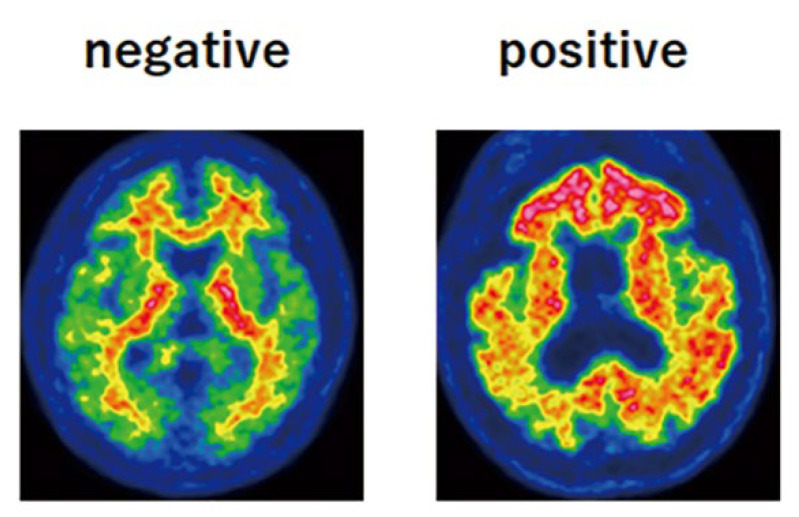
The evaluation was performed according to the criteria defined by the manufacturer, which involved visual assessment of five regions: frontal cortex, PC/PCC, lateral-parietal, lateral temporal, and striatum. The final classification was either positive (unilateral binding in one or more cortical brain regions or striatum; (**right**)) or negative (predominantly white matter uptake; (**left**)).

**Figure 5 jcm-11-06118-f005:**
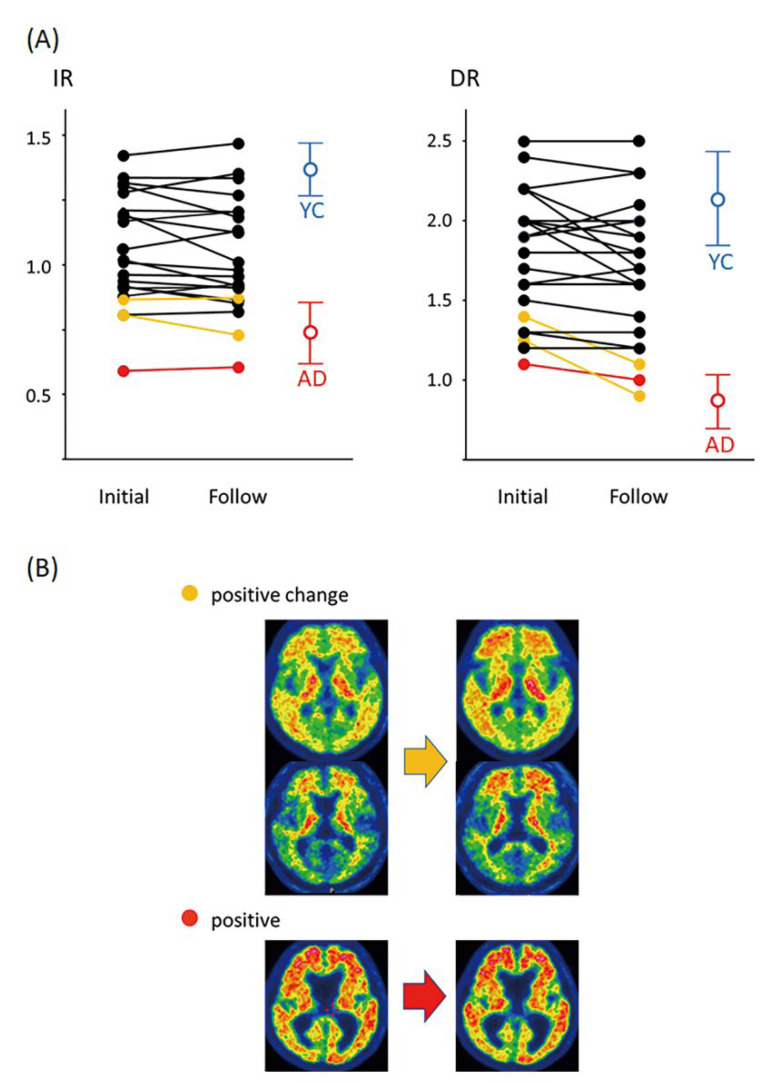
Relationship between intentional flow and β-amyloid accumulation. (**A**) Changes in IR and DR over 2 years. In the [^15^O]H_2_O PET examinations conducted at the start and after two years, IR was 1.03 ± 0.21 and 1.02 ± 0.20 and DR was 1.74 ± 0.43 and 1.67 ± 0.47, respectively, with both indices showing no significant changes over the 2-year period. The blue and red open circles and bars represented the mean and standard deviation of the results for young controls (YC) and AD patients (AD), respectively (IR: 1.38 ± 0.08 and 0.73 ± 0.09; DR: 2.12 ± 0.32 and 0.86 ± 0.17). Black dot: negative β-amyloid results on initial and follow-up PET examinations. Yellow dot: positive β-amyloid changes on follow-up PET examinations. Red dot: positive β-amyloid results on initial and follow-up PET examinations. (**B**) Changes in β-amyloid accumulation over 2 years. Upper: PET images showing a positive change (2 participants). Lower: PET images showing positive findings (1 participants).

**Table 1 jcm-11-06118-t001:** Follow-up study summary.

	Initial Examination	Follow-Up Examination
Age (years)	67.68 ± 6.31	69.98 ± 6.25
Influx ratio (IR)	1.03 ± 0.21	1.02 ± 0.20
Drain rate (DR)	1.74 ± 0.43	1.67 ± 0.47

## Data Availability

Not applicable.

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
