# Peer review of "Blood Cerebrospinal Fluid Barrier Function Disturbance Can Be Followed by Amyloid-β Accumulation"

_jcm, 2022, doi:10.3390/jcm11206118_

Round 1

Reviewer 1 Report

This paper reports a two-year study to correlate water flow from cerebral cortex to ventricular CSF (assessed by 15O-H2O PET) with amyloid-beta aggregation (assessed by 18F-flutemetamol PET).  The authors find that water flow is lower in amyloid-beta-positive subjects than in amyloid-beta-negative subjects, and may decrease as amyloid-beta aggregation increases.  The number of subjects is small (only one subject with initial amyloid-beta positivity and two subjects who developed positivity over two years).  However, the authors describe their methods and their results thoroughly and conservatively.  They do not make unjustified claims.  

Reviewer 2 Report

In this interesting article authors show that disturbances in the proper clearance of Aβ by interstitial flow into the CSF can play a significant role in senile plaque formation and ultimately the pathogenesis of AD. This study is an exploratory study and sample size is very small. The major concern for me is its smaller sample size and observation period for determination of the relationship between interstitial flow and Aβ accumulation. At the same time I see a strong conclusion based on concrete evidence. Manuscript is written nicely and data were presented clearly. I expect authors follow this study further without discontinuation.

Reviewer 3 Report

Review of article of Suzuki et al. on:

Blood cerebrospinal fluid barrier function disturbance can be 2 followed by amyloid-β accumulation.

In the article the authors tested the Cerebral spinal in and outflow of water with Radioactive 15O labelled water. In addition, the authors measured Beta Amyloid concentration in the brain using 18F flutemetamol.

The topic of the article could be of interest however, the data presented in the article are far from complete to be able to make the point.

The analysis of influx and drainage of the water content in the brain is an interesting method, specifically as there is a 2 year interval to measure the same people. There are only 2 people that started developing beta amyloid plaques in this period, and one already had plaques from the start. The other people in the study did not have plaque development I assume.

Why not show the 18F flutemetamol PET scanning of all people in the study?

It is difficult to see if there is an increase in 18F flutemetamol PET signal in the 2 people that should have developed plaques. There is no analysis of the staining intensity of these people/patients. 

The in flow look like for these 3 selected people in the study to go down, but there are others that are going down more or are at least as low or lower as some of the 2 selected people but according to the authors did not developed plaques. 

Concerning the drainage the 3 selected are lower, but some are almost as low (not significantly different from the selected) and did not have 18F flutemetamol increased signal. 

Therefore, there is no data to support the claims of the article and part of the data is missing.

The hypothesis of AQP4 being involved, could be, but there is no data supporting this.

Round 2

Reviewer 3 Report

I appreciate that the authors have used quite some work on this study, but there is no significant effect of the data presented to support the hypothesis. 

This is a pure hypothesis that can be true but can also be false.